# The Role of New Morphological Parameters Provided by the BC 6800 Plus Analyzer in the Early Diagnosis of Sepsis

**DOI:** 10.3390/diagnostics14030340

**Published:** 2024-02-04

**Authors:** Sara Sacchetti, Matteo Vidali, Teresa Esposito, Stefano Zorzi, Alessia Burgener, Lorenzo Ciccarello, Gianmaria Cammarota, Valentina Zanotti, Luca Giacomini, Mattia Bellan, Mario Pirisi, Ramon Simon Lopez, Umberto Dianzani, Rosanna Vaschetto, Roberta Rolla

**Affiliations:** 1Clinical Chemistry Laboratory, Department of Health Sciences, Università del Piemonte Orientale, Maggiore della Carità University Hospital, 28100 Novara, Italy; sara.sacchetti@uniupo.it (S.S.); valentina.zanotti@uniupo.it (V.Z.); 20023552@studenti.uniupo.it (L.G.); umberto.dianzani@med.uniupo.it (U.D.); roberta.rolla@med.uniupo.it (R.R.); 2Clinical Pathology Unit, Foundation IRCCS Ca’ Granda Ospedale Maggiore Policlinico, 20122 Milan, Italy; 3Unit of Anaesthesia and Intensive Care, Department of Translational Medicine, Università del Piemonte Orientale, Maggiore della Carità University Hospital, 28100 Novara, Italy; expoterry@gmail.com (T.E.); stefano.zorzi.pr@gmail.com (S.Z.); alessia.burgener@gmail.com (A.B.); lorenzo.ciccarello@outlook.it (L.C.); gianmaria.cammarota@uniupo.it (G.C.); rosanna.vaschetto@med.uniupo.it (R.V.); 4Department of Translational Medicine, Division of Internal Medicine, Università del Piemonte Orientale, “Maggiore della Carità” University Hospital, 28100 Novara, Italy; mattia.bellan@med.uniupo.it (M.B.); mario.pirisi@med.uniupo.it (M.P.); 5Medical Xpert Systems SA, Brunnenmattstrasse 6, 6317 Oberwil bei Zug, Switzerland; ramonsimonlopez@gmail.com

**Keywords:** affordable health care, early diagnosis, leukocyte parameters, morphological changes, Red Cell Distribution Width (RDW), sepsis

## Abstract

Background: Late diagnosis of sepsis is associated with adverse consequences and high mortality rate. The aim of this study was to evaluate the diagnostic value of hematologic research parameters, that reflect the cell morphology of blood cells, available on the BC 6800 plus automated analyzer (Mindray) for the early detection of sepsis. Materials and Methods: A complete blood count (CBC) was performed by Mindray BC 6800 Plus Analyzer in 327 patients (223 with a confirmed diagnosis of sepsis following sepsis-3 criteria, 104 without sepsis), admitted at the Intensive Care Unit of the Novara’s Hospital (Italy) and in 56 patients with localized infection. Results: In univariate logistic regression, age, Hb, RDW, MO#, NMR, NeuX, NeuY, NeuZ, LymX, MonX, MonY, MonZ were associated with sepsis (*p* < 0.005). In multivariate analysis, only RDW, NeuX, NeuY, NeuZ, MonX and MonZ were found to be independent predictors of sepsis (*p* < 0.005). Morphological research parameters are confirmed to be predictors of sepsis even when analyzing the group with localized infection. Conclusions: In addition to already established biomarkers and basic CBC parameters, new morphological cell parameters can be a valuable aid in the early diagnosis of sepsis at no additional cost.

## 1. Introduction

Sepsis is a life-threatening condition caused by a systemic and dysregulated host response to infection [1]. Sepsis is one of the leading causes of death in hospitalized patients and is also a major contributor to neonatal mortality and morbidity, especially in low- and middle-income countries [2,3]. According to the World Health Organization (WHO) and the Global Burden of Diseases report, sepsis is estimated to affect nearly 50 million people each year, causing 11 million deaths annually, accounting for 20% of all deaths worldwide [4]. Initiatives such as the World Sepsis Alliance and World Sepsis Day have raised awareness of sepsis and promote sepsis prevention and early detection, which are critical to improving treatment outcomes [5,6].

The diagnosis of sepsis is based on clinical and laboratory parameters that indicate the presence of infection and organ dysfunction. The most specific test for the diagnosis of sepsis is blood cultures, which are used to confirm the diagnosis of sepsis and identify the source of infection. However, their sensitivity is quite low, and the results are usually only available about 2 days after sampling [7]. Due to this time delay, alternative circulating biomarkers with higher sensitivity and availability of results are commonly used. Currently, the most important markers for sepsis diagnosis are procalcitonin (PCT) and C-reactive protein (CRP), which reflect the severity of the infection and can help in treatment decisions [8,9]. However, CRP has a significant drawback as its specificity is limited; it can also be elevated in various other inflammation-related diseases [10]. PCT, on the other hand, is considered one of the most specific markers for bacterial sepsis, and its concentration is related to disease severity, mortality, and organ failure. However, PCT may give to false negative results in patients with invasive fungal infections and may be elevated in non-infectious diseases. In addition, the cost of PCT tests is relatively high, making them unattractive for routine use [10]. Therefore, serial PCT tests are recommended primarily for surveillance purposes and to guide appropriate antibiotic treatment rather than as a first-line diagnostic tool [8].

Laboratory medicine holds a pivotal role in sepsis diagnosis, continuously exploring novel biomarkers to enhance diagnostic capabilities. Among these prospects, CD64, also referred to as Fc receptor I, stands out as a rising biomarker. This high-affinity receptor on neutrophils binds to the Fc portion of the Immunoglobulin-G heavy chain. Neutrophil CD64 orchestrates functions within both innate and adaptive immune responses, and its levels notably surge within 4–6 h in response to infection-induced proinflammatory cytokines. The neutrophil expression of CD64 emerges as a promising candidate biomarker for sepsis [11].

In addition, several studies have shown that the morphological parameters of leukocytes play an important role in the diagnosis of sepsis [12]. One example is the monocyte distribution width (MDW), a new laboratory parameter obtained from a complete blood count (CBC) that measures the morphological and dimensional variability of monocytes in the blood [13,14]. The MDW is a promising tool for the diagnosis and treatment of sepsis, and its use in clinical practice can improve outcomes in sepsis patients [15,16]. However, the MDW parameter is only calculated using new-generation hematology analyzers from Beckman Coulter (Miami, Florida, USA), which limits its wide use [17].

Several other inexpensive and easily accessible research-use-only (RUO) cell parameters can be obtained from blood count results. In particular, the BC-6800 Plus Analyzer (Mindray, China) offers a high-resolution, three-dimensional SF cube technology that provides detailed information on blood cell morphology in terms of leukocyte volume, cytoplasmic and nuclear complexity, and nucleic acid content (DNA and RNA). This information is automatically converted into quantitative RUO parameters that describe the morphology of the cells: NeuX, NeuY, NeuZ for neutrophils; MonX, MonY, MonZ for monocytes; and LymX, LymY, LymZ for lymphocytes.

The aim of the present study was to evaluate the diagnostic value of these RUO blood cell parameters provided by the BC 6800 Plus Analyzer for the early detection of sepsis in the intensive care unit (ICU).

## 2. Materials and Methods

### 2.1. Study Design

A retrospective observational study was conducted at the University Hospital (AOU) “Maggiore della Carità”, Novara, Italy. The study protocol was approved by the Ethical Committee of the University Hospital “Maggiore della Carità” (CE 127/2023), Novara, Italy, and performed in accordance with the current revision of the Helsinki Declaration.

### 2.2. Study Population

Laboratory data from day 1, corresponding to ICU admission, were analyzed. Data were extracted from the Laboratory Information System (LIS) into an Excel database using TD Synergy software 12.21 (Siemens Healthcare, Erlangen, Germany) and LabExpert software (Mindray, Shenzhen, China) and were anonymized.

We reviewed the electronic medical records of 223 patients admitted to the ICU between July 2021 and October 2022 who were subsequently diagnosed with sepsis. The criteria for sepsis diagnosis were based on the Third International Consensus Definitions of Sepsis and Septic Shock (Sepsis-3 Criteria) [1]. The consensus recommends a sepsis diagnosis with suspected infection and a Sequential Organ Failure Assessment (SOFA) score (used to determine the level of organ dysfunction and mortality risk in ICU) of 2 or higher [18]. Only patients with suspected infection and a SOFA score ≥ 2 but negative for SARS-CoV-2 were selected.

A control group of 104 individuals consecutively admitted to the ICU for a traumatic and non-infectious event was also included in the study. Exclusion criteria were missing quantitative RUO parameters, underlying hemato-oncological disease and positivity for SARS-CoV-2.

Demographic characteristics and SOFA score at ICU admission were collected for all patients. As components of the SOFA score, the Glasgow Coma Scale (GCS) [19], the ratio of arterial oxygen partial pressure to fractional inspired oxygen (PaO_2_/FiO_2_), mean arterial pressure (MAP), blood creatinine, bilirubin levels and blood culture test results were recorded. Furthermore, blood lactates, ICU and hospital length of stay (LOS) and mortality were evaluated.

Then, a third group of 56 patients with urinary tract infections admitted to the Internal Medicine Unit was subsequently recruited in order to validate the diagnostic model of sepsis obtained from the first two groups of patients. To do so, we reviewed all the clinical records of the patients admitted to the ward from 1 July 2021 to 1 July 2023, and we selected all those with a documented respiratory or urinary infection and a SOFA Score < 2. The exclusion criteria were missing quantitative RUO parameters, underlying hemato-oncological disease and positivity for SARS-CoV-2.

### 2.3. Data Collection

CBC was performed on EDTA-K2 anticoagulant whole blood samples using the BC-6800 Plus Hematology Analyzer (Mindray, Shenzhen, China); CRP was measured on lithium heparin anticoagulant blood samples using the ADVIA 1800 Clinical Chemistry System (Siemens Healthineers, Erlangen, Germany); and PCT was determined on lithium heparin anticoagulant blood samples using the ADVIA Centaur^®^ XP Immunoassay System (Siemens Healthiness, Erlangen, Germany).

The quantitative RUO parameters NeuX, NeuY, NeuZ, MonX, MonY, MonZ, LymX, LymY, LymZ provided by the BC 6800 Plus Analyzer were assessed. These parameters are numerical data indicating the center of gravity position of the measured events. They are related to cell morphology (volume, granularity, and complexity of each cell) and are useful for evaluating and characterizing the cells in a blood sample [20,21].

In particular, NeuX, MonX and LymX represent the cytoplasmic granularity and/or complexity of the cytoplasm and nucleus of neutrophils, monocytes and lymphocytes. A high NeuX value indicates increased granularity and is a sign of neutrophil activation and/or abnormality. High MonX and LymX values indicate increased cytoplasmic and nuclear complexity and are indicative of activation and/or cellular abnormalities.

NeuY, MonY and LymY measure the light scattering properties of neutrophils, monocytes and lymphocytes. A high Y value (high cellular fluorescence) indicates cell activation and is due to increased absorption of fluorochrome in the nucleus due to less thickened chromatin and in the cytoplasm due to increased amounts of mRNA supporting increased protein synthesis.

NeuZ, MonZ and LymZ are parameters that reflect the cell sizes. They provide an estimate of cell size and can be used to assess changes in the morphology of neutrophils, monocytes and lymphocytes. Deviations from the normal range of the Z value may indicate cell activation and/or abnormalities.

The stability of the cellular RUO position parameters was first analyzed on a group of 80 consecutive random samples from routine procedures. All samples were processed within two hours of blood collection. The experimental data (study design: 80 samples, stored at room temperature or at 4 °C and measured a t_0_, t_1_ = 2 h, t_2_ = 8 h, t_3_ = 24 h) show that all evaluated quantitative RUO parameters are stable up to 2 h after blood collection (deviations < 4%). Storing the sample at 4 °C did not provide any advantage. In addition, the repeatability of the cellular RUO parameters was tested by analyzing 50 samples with 3 different modules of the BC 6800 Plus Analyzer. The repeatability for all cellular parameters was <4%.

The Neutrophil-to-Monocyte Ratio (NMR), Lymphocyte-to-Monocyte Ratio (LMR), and Red Blood Cell Distribution Width (RDW) were also examined as prognostic markers for adverse outcomes in sepsis and as predictors of bacterial infection [22,23,24,25].

RDW is an erythrocyte index that measures the variation in the size of RBCs in a blood sample. It expresses the dispersion or distribution width of RBCs around their mean corpuscular volume (MCV). RDW is usually expressed as a percentage and is a measure of the heterogeneity or anisocytosis of red blood cells. A high RDW value indicates greater variability in RBC sizes, while a low value indicates uniform sizes.

### 2.4. Statistical Analysis

Statistical analyses were performed using SPSS statistical software v.17.0 (SPSS Inc., Chicago, IL, USA) and R Language v.4.3.1 (R Foundation for Statistical Computing, Vienna, Austria). Normality distribution was assessed preliminarily using q-q plot and Shapiro–Wilk tests. Differences between the groups for continuous variables or categorical were estimated via Mann–Whitney U test and Fisher’s exact test, respectively. Diagnostic accuracy for the prediction of sepsis was evaluated via Receiver Operating Characteristic (ROC) curve analysis and reported as the Area Under the Curve (AUC) and 95% confidence interval. Differences between the AUCs were evaluated using the DeLong method. Differences in mortality were evaluated using the Log Rank test.

The association between predictors and sepsis was examined using univariate and multivariate logistic regression. The association between predictors and time to mortality within the ICU or hospital was evaluated using univariate and multivariate Cox regression.

## 3. Results

A total of 327 electronic records were reviewed, of which 223 patients had sepsis (60% males, 40% females), while 104 patients (68% males, 32% females) were not septic and were used as controls. The population characteristics are shown in Table 1. Overall, patients had a median age of 70 (57–77) years, with an average SOFA score of 6 (4–8) and a Glasgow Coma Score (GCS) at ICU admission of 15 (9–15). Age, mean arterial pressure (MAP) and the ratio of arterial oxygen partial pressure to fractional inspired oxygen (PaO_2_/FiO_2_) on admission differed significantly between patients with and without sepsis (71 (60–78) vs. 64 (52–75) years, 79 (65–93) vs. 90 (71–107) mmHg, and 175 (116–274) vs. 303 (174–403) mmHg, respectively, *p* < 0.05), whereas median SOFA score was not dissimilar between the two groups. Furthermore, levels of creatinine (114.95 (61.89–203.37) vs. 61.89 (53.05–88.42) μmol/L, *p* < 0.05) and lactate (1.6 (1.0–3.1) vs. 1.4 (0.9–2.6) mmol/L) levels were higher in the sepsis group compared to the controls, while no significant differences were found in terms of bilirubin concentrations. Bacteremia was suspected in 81% of the patients. Blood cultures were positive in 36% of the cases. Coagulase-negative Staphylococcus was found in 18 patients. Lastly, ICU and hospital length of stay differ between the two groups (respectively, *p* = 0.05 and *p* = 0.011), and mortality was significantly higher in sepsis patients than in the controls (ICU mortality 34% vs. 20%, Log Rank test *p* = 0.002; hospital mortality 45% vs. 21%, Log Rank test *p* = 0.012).

Initially, only blood counts were analyzed to assess the ability of blood count parameters alone, both standard and RUO, for the early diagnosis of sepsis. On admission, many hematological parameters differed significantly between patients with and without sepsis, as detailed in Table 2: hemoglobin (Hb), Red Cell Distribution Width (RDW), White Blood Cell Count (WBC), lymphocyte count (LY#), monocyte count (MO#), neutrophil-to-monocyte ratio (NMR), lymphocyte-to-monocyte ratio (LMR), NeuX, NeuY, NeuZ, LymX, MonX, MonY, MonZ, CRP and PCT.

In univariate logistic regression analysis, age, Hb, RDW, MO#, NMR, NeuX, NeuY, NeuZ, LymX, MonX, MonY and MonZ were associated with sepsis (Table 3). However, in multivariate analysis, only RDW, NeuX, NeuY, NeuZ, MonX and MonZ were found to be independent predictors of sepsis. The multivariate model correctly classified up to 85% of cases (false negative FN 7%, false positive FP 7%). The AUC for sepsis of the multivariate model was 0.92 (95%CI 0.89–0.95) (Figure 1).

In addition, CRP and PCT were significantly elevated on admission in patients with sepsis compared with patients without sepsis (Table 2). The AUCs for sepsis of CRP and PCT were 0.83 (95%CI 0.79–0.88; *p* < 0.001) and 0.78 (95%CI 0.73–0.84; *p* < 0.001), respectively (Table 4). However, the AUC for sepsis of our multivariate model was significantly higher than the AUCs of both CRP and PCT (Bonferroni’s correction, both *p* < 0.001). No difference was evident between CRP and PCT. Finally, CRP was added to the univariate and multivariate analysis: the new model correctly classified up to 88% of cases (FN 7%, FP 5%).

Predictors of mortality within the ICU and within a hospital were evaluated via univariate and multivariate Cox regression. As detailed in Table 5, at the univariate Cox regression analysis, age, RDW, NeuX, NeuY, LymY, LymZ and MonX were associated with mortality within the ICU, whereas age, RDW, NLR, NeuX, NeuY, LymY, LymZ, MonX and MonY were associated with mortality within a hospital. At the multivariate Cox regression analysis, age, NeuY, LymY and MonX were independent predictors of shorter time to mortality within ICU, whereas age, NeuY, LymY and MonY were independent predictors of shorter time to mortality within a hospital (Table 5).

Finally, the third group of 56 patients with localized infection without sepsis (SOFA Score < 2) was compared to the sepsis group, and it was comparable in terms of gender distribution (M:F 27:29 vs. 134:89; *p* = 0.13) but significantly older (82 (76–90) vs. 71 (60–78) *p* < 0.001).

Several hematological parameters of the patients with localized infection differed significantly from those of patients with sepsis, as shown in Table 6. In univariate analysis, age, Hb, MO#, NLR, NMR, PLT, NeuX, NeuY, MonX, MonY and CRP were found to be associated with sepsis (Table 7). As detailed in Table 7, in multivariate analysis without and with CRP, only age, Hb, NMR, PLT, NeuY and MonY were found to be independent predictors of sepsis. The multivariate model correctly classified up to 89% of cases (false negative FN 4%, false positive FP 7%). The AUC for sepsis of the multivariate model was 0.91 (95%CI 0.86–0.95) (Figure 2). In addition, CRP and PCT were significantly elevated on admission in patients with sepsis compared with patients without sepsis (Table 6). The AUCs for sepsis of CRP and PCT were 0.62 (95%CI 0.54–0.69; *p* = 0.008) and 0.69 (95%CI 0.59–0.78; *p* < 0.001), respectively (Figure 2). However, the AUC for sepsis of our multivariate model was significantly higher than the AUCs of both CRP and PCT (Bonferroni’s correction, both *p* < 0.001). No difference was evident between CRP and PCT. Finally, CRP was added to the univariate and multivariate analysis: the new model correctly classified up to 89% of cases (FN 4%, FP 7%).

## 4. Discussion

In recent decades, several new early biomarkers for sepsis have been tested, such as matrix metalloproteinase-9 (MMP-9), its endogenous inhibitor tissue inhibitor of metalloproteinase-1 (TIMP-1), and mid-regional pro-adrenomedullin (MR-proADM). MMP-9 and TIMP-1 are key regulators of inflammation, and disturbances in their dynamic balance of expression and activity may contribute to tissue damage and increased mortality in sepsis. MR-proADM is measured in place of adrenomedullin (ADM) because it is rapidly cleared from circulation. ADM plays an important role in inflammation and progression from sepsis to septic shock, and MR-proADM appears to be a promising alternative to the Sequential Organ Failure Assessment (SOFA) score for the assessment of organ failure and prediction of mortality in septic patients [26].

Unfortunately, few of these new biomarkers for early sepsis are currently used in clinical practice. This limitation is due to factors such as cost, availability and problems associated with low sensitivity or specificity. The complexity of patients further complicates research in this area and requires concerted actions to overcome these challenges through a combination of biomarkers.

In recent years, several studies have emphasized the central role of complete blood count (CBC) analysis and morphological parameters in the early diagnosis of sepsis. Buoro et al. emphasized that parameters such as the immature platelet fraction (IPF#) and the reticulocyte ratio (RET %) are risk predictors for the development of sepsis in critically ill patients and allow early patient management before clinically visible systemic infections occur [27]. Pelagalli et al. presented a study showing that leukocyte differential count and several parameters related to cell population data (CPD) determined with the Mindray BC-6800 Plus analyzer have predictive value for sepsis screening. This suggests that blood count and CPD can be used as plausible markers for early sepsis diagnosis in emergency situations, facilitating timely clinical intervention [28].

The present study focused on the analysis of several standard and RUO cellular morphological parameters derived from patient CBC results, which are accessible and cost-effective for all laboratories, not only for users of the BC 6800 plus. The multivariate analysis identified NeuX, NeuY, NeuZ, MonX, MonZ and RDW as independent predictors of sepsis. Combining these parameters in a multivariate model showed strong predictive power and correctly classified up to 85% of cases. NeuX, NeuY, NeuZ, MonX, MonZ and RDW proved to be useful parameters for early diagnosis of sepsis and performed better than using PCT and/or CRP alone. The multivariate analysis model was improved with the addition of CRP and was able to correctly classify up to 88% of cases. PCT was not included in the statistical model because it is a costly test with a high rate of inappropriate queries and should be used primarily to guide antibiotic therapy. In addition, NeuY, MonX and LymY were found to be independent predictors of shorter time to mortality within ICU/hospital, which is also useful for clinicians in early prediction of the outcome of sepsis. The RUO parameters could, therefore, also be useful in determining appropriate therapeutic strategies for septic patients.

The importance of neutrophil and monocyte morphological parameters in sepsis has already been underlined in previous studies and is in line with the role of these cells in acute inflammation and defense from bacterial infections [21,29,30]. Accurate measurement of these morphologic changes in septic patients undoubtedly provides additional value and clinical significance to the routine indices of the complete blood count (CBC). This is particularly important in situations where monitoring other sepsis biomarkers is too costly, as is often the case in low-income countries and in neonates [31]. By contrast, it is not surprising that our analysis did not highlight lymphocyte parameters since changes in the morphology of these cells are mainly expected to mark viral infections.

Our study confirmed in a Caucasian population the data presented by Zhang et al., who showed that the parameters NeuX, NeuY, MonX and MonY determined by the BC 6800 Plus analyzer (Mindray) were significantly increased in the sepsis group compared with the bacterial infection group, indicating their potential usefulness in sepsis diagnosis and in differentiating bacterial infections. In addition, our study confirmed that the morphologic parameters, especially NeuY-, have higher diagnostic predictive power for sepsis compared to CRP [21]. Under the action of various stimuli, monocytes undergo morphological and structural changes, which reflect their activation state and allow them to perform multiple biological functions, modulating the immune response and the resolution and/or progression of the disease [32]. Changes in the volumetric dimensions of monocytes may be ascribed to their activation, associated with phagocytosis of pathogens and acquirement of features of large amoeboid cells. This transformation is accompanied by increased expression of functional markers, such as CD16 [16]. The activation of monocytes leads to an increase in monocyte size, which is quantified using the MonZ parameter in the blood count, while the changes in monocyte structure are detected using the MonX parameter.

Similarly, neutrophils play a critical role as effectors of the innate immune response against bacterial infections, and NeuX, NeuY and NeuZ can be helpful in assessing the neutrophil morphologic features and detecting abnormalities or deviations from the normal state [33].

The main function of neutrophils is to eliminate pathogens through phagocytosis, a process in which they engulf and destroy bacteria, fungi and other microorganisms, effectively eliminating infections [34]. In addition, neutrophils release antimicrobial peptides and reactive oxygen species that further enhance their antimicrobial activity and contribute to the immune response against invading pathogens [34]. In sepsis, the normal function of neutrophils may be impaired, leading to dysregulation and potentially harmful effects. The immune response can become dysregulated, leading to an overwhelming inflammatory response known as a cytokine storm [35]. In this state, neutrophils can become hyperactive, leading to an excessive and uncontrolled release of proinflammatory molecules. These include cytokines such as tumor necrosis factor-alpha (TNF-α), interleukin-1 (IL -1) and IL -6, which can contribute to tissue damage, organ dysfunction and systemic inflammation [36]. In sepsis, neutrophils may be functionally impaired, with reduced phagocytosis capacity, impaired chemotaxis and inadequate ability to kill microorganisms. These dysfunctions may impair the ability of neutrophils to effectively clear infections and contribute to the persistence and spread of pathogens [36] with deleterious effects on the host. The excessive release of proinflammatory cytokines and the inability to clear the infection can lead to extensive tissue damage, organ dysfunction and systemic inflammatory response syndrome (SIRS), which is a hallmark of severe sepsis [37].

In addition, neutrophils can contribute to hyperinflammation in sepsis by releasing neutrophil extracellular traps (NETs). These traps consist of DNA strands decorated with antimicrobial proteins such as histones and granular enzymes that can intercept and kill pathogens [38]. During sepsis, neutrophils may undergo netosis as part of their response to infection. When activated, neutrophils release NETs into the environment to ensnare and neutralize invading microorganisms [39]. NETs can immobilize bacteria, fungi and other pathogens, preventing their spread and promoting their elimination. While netosis serves as an important defense mechanism against pathogens, it can also have detrimental effects on sepsis [39]. Excessive NET formation can lead to clot formation, clogging blood vessels, impairing blood flow, and potentially causing tissue damage. In addition, the release of NETs may contribute to the inflammatory response in sepsis by triggering the production of proinflammatory cytokines and amplifying the systemic inflammatory cascade [39].

Finally, in our study, the morphological parameter of red blood cells, RDW, also proved to be an early independent predictor of sepsis risk in ICU patients. It has already been established that there is a correlation between the onset and development of sepsis and the level of RDW, which is an important predictor of death in patients with sepsis [24,25], and is closely related to inflammatory markers such as CRP. It has been postulated that inflammation may impair the maturation process of red blood cells, leading to accelerated entry of immature erythrocytes into the circulation [40]. Moreover, RDW is influenced by the lifespan of erythrocytes and has been suggested to be an indicator of a chronic inflammatory response, in contrast to conventional markers like CRP that primarily reflect an ongoing, acute inflammatory reaction. In addition, an increased RDW value in sepsis patients could indicate the presence of organ dysfunction and, consequently, a poorer prognosis [25].

Due to its retrospective nature, our study has a limitation that needs to be discussed. Patients with confirmed sepsis were compared with an ICU population affected by non-septic disease, whereas a more appropriate control group would have consisted of patients with suspected but unproven sepsis. Such categorization is often not possible when retrospectively recruiting patients by analyzing electronic records. Furthermore, according to clinical guidelines [41], ICU patients with a strong suspicion of sepsis are treated as septic regardless of confirmation of infection. To overcome this limitation, a third control group with localized infection but without sepsis was recruited. In the univariate logistic regression analysis, the cellular RUO position parameters NeuX, NeuY, MonX and MonY were associated with sepsis. The multivariate analysis specifically identified NeuY and MonY as independent predictors of sepsis, confirming the role of RUO parameters in the early diagnosis of sepsis. The combination of these parameters in a multivariate model showed strong predictive power and correctly classified up to 89% of cases.

In conclusion, the integration of morphologic research parameters from automated blood cell analyzers together with traditional clinical signs, the SOFA score and established biomarkers could improve sepsis diagnosis and treatment, facilitate rapid intervention and potentially lead to better patient outcomes. RUO cell parameters can be accurately measured with any new-generation hematology analyzer and provide additional information to conventionally reported blood count parameters. Importantly, this additional analysis does not incur additional costs compared to a normal blood count, making RUO testing cost-effective and accessible to all laboratories, including non-specialized spokes laboratories. Consequently, the RUO parameters could be used worldwide for the early diagnosis of sepsis at no additional cost.

Despite the limitations discussed, the data presented emphasize the diagnostic potential of the integrated use of RUO cell parameters, which accurately reflect the morphological and functional changes of leukocytes and erythrocytes during sepsis and beyond, in combination with the parameters already recorded in the complete blood count. The use of morphological RUO parameters is very promising for clinical practice. and their future use could prove crucial for the development of diagnostic algorithms for different pathologies thanks to the combined use of Artificial Intelligence and Machine Learning. However, further validation of the available data in other cohorts, in larger studies and in a prospective study, is needed. Future studies should focus on investigating the diagnostic utility and potential clinical applications of morphologic parameters to improve sepsis outcomes and patient care.

## Figures and Tables

**Figure 1 diagnostics-14-00340-f001:**
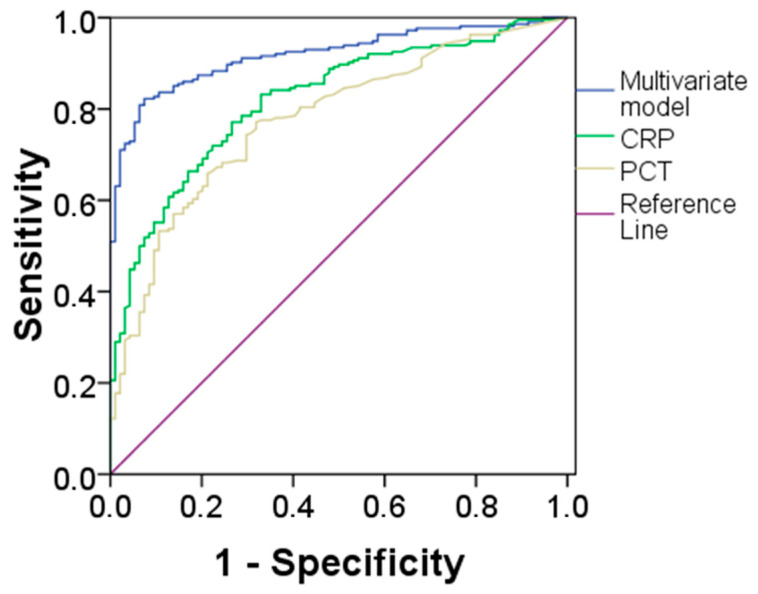
ROC curve analysis for the comparison between multivariate model, CRP and PCT in sepsis prediction. CRP, C-reactive protein; PCT, procalcitonin.

**Figure 2 diagnostics-14-00340-f002:**
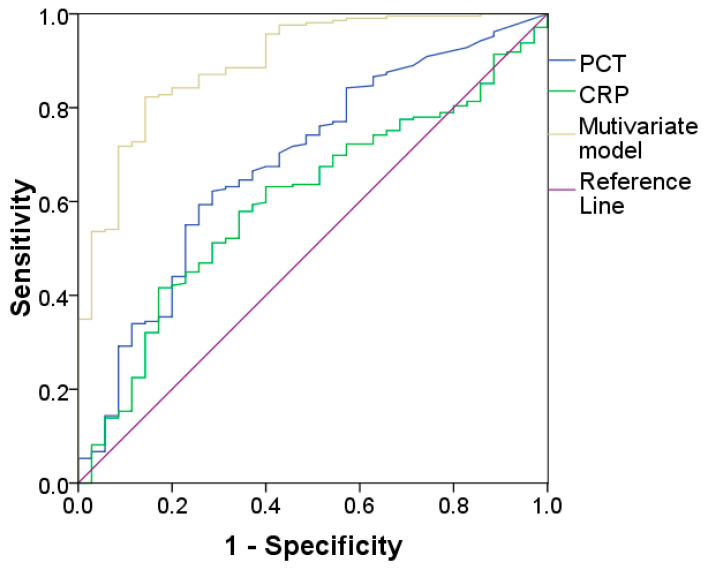
ROC curve analysis for the comparison between multivariate model, CRP and PCT in sepsis prediction. CRP, C-reactive protein; PCT, procalcitonin.

**Table 1 diagnostics-14-00340-t001:** Patient demographic and clinical characteristics.

Demographic and Clinical Characteristics	All Patients *n* = 327	Sepsis Patients *n* = 223	Non-Sepsis Patients *n* = 104
**Age, years**	70 (57–77)	**71 (60–78)**	**64 (52–75)**
**Male, *n* (%)**	205 (63)	134 (60)	71 (68)
**Female, *n* (%)**	122 (37)	89 (40)	33 (32)
**SOFA SCORE**	6 (4–8)	6 (4–8)	6 (4–7)
**PaO_2_/FiO_2_, mmHg**	209 (130–332)	**175 (116–274)**	**303 (174–403)**
**PLT, ×10^9^/L**	195 (131–259)	190 (116–261)	206 (163–255)
**MAP, mmHg**	81 (67–97)	**79 (65–93)**	**90 (71–107)**
**Bilirubin, μmol/L**	13.68 (10.26–23.94)	15.39 (10.26–25.65)	13.68 (8.55–20.52)
**Creatinine, μmol/L**	87.54 (61.89–150.31)	**114.95 (61.89–203.37)**	**61.89 (53.05–88.42)**
**GCS**	15 (9–15)	**15 (14–15)**	**8 (5–14)**
**Lac, mmol/L**	1.6 (1–3)	**1.6 (1.0–3.1)**	**1.4 (0.9–2.6)**
**ICU LOS, d**	3 (1–9)	3 (1–7)	5.5 (1.0–12)
**Hospital LOS, d**	17 (8–32)	**19 (10–33)**	**11 (6–28)**
**ICU mortality, n (%)**	98 (30)	**77 (34)**	**21 (20)**
**Hospital mortality, n (%)**	122 (37)	**100 (45)**	**22 (21)**

Data are presented as numbers and percentages or median and IQR. SOFA: Sequential Organ Failure Assessment; PaO_2_/FiO_2_: ratio of arterial oxygen partial pressure to fractional inspired oxygen; PLT: platelets; GCS, Glasgow Coma Score; Lac: lactate; MAP: mean arterial pressure; ICU, intensive care unit; LOS, length of stay. Bold values denote statistical significance at the *p* < 0.05 level.

**Table 2 diagnostics-14-00340-t002:** Hematologic and biochemical data of the study population divided into sepsis and without sepsis groups.

Predictor	Without Sepsis (*n* = 104)	With Sepsis (*n* = 223)	*p*-Value
**Hb, g/L**	127 (107–142)	108 (90–128)	**<0.001**
**RDW, %**	13.4 (12.7–14.3)	14.8 (13.8–16.6)	**<0.001**
**WBC, ×10^9^/L**	12.7 (10.1–16.1)	11.2 (6.8–15.8)	**0.022**
**NE#, ×10^9^** **/L**	11.0 (8.1–14.1)	9.6 (5.5–14.0)	0.061
**LY#, ×10^9^/L**	0.8 (0.5–1.3)	0.6 (0.4–1.1)	**0.002**
**MO#, ×10^9^/L**	0.6 (0.4–0.9)	0.4 (0.2–0.7)	**<0.001**
**NLR**	13.6 (6.6–22.5)	13.4 (6.9–24.5)	0.644
**NMR**	16.8 (12.2–24.3)	21.5 (13.3–36.2)	**<0.001**
**LMR**	1.3 (0.8–2.1	1.5 (0.9–2.8)	**0.030**
**PLT, ×10^9^/L**	206 (163–255)	190 (116–261)	0.099
**NeuX**	361 (345–389)	408 (371–446)	**<0.001**
**NeuY**	479 (455–500)	541 (495–607)	**<0.001**
**NeuZ**	1858 (1793–1910)	1792 (1712–1874)	**<0.001**
**LymX**	94 (90–99)	97 (91–104)	**0.002**
**LymY**	765 (736–805)	775 (728–833)	0.203
**LymZ**	962 (944–978)	954 (931–982)	0.484
**MonX**	208 (202–218)	224 (211–245)	**<0.001**
**MonY**	1046 (996–1080)	1144 (1065–1225)	**<0.001**
**MonZ**	1312 (1292–1334)	1348 (1303–1408)	**<0.001**
**CRP, mg/L**	20.6 (8.6–54.3)	140.6 (64.8–207.2)	**<0.001**
**PCT, ng/mL**	0.17 (0.08–0.54)	2.67 (0.36–19.60)	**<0.001**

Hb, hemoglobin; RDW, red distribution width; WBC, white blood cells; NE#, absolute number of neutrophils; LY#, absolute number of lymphocytes; MO#, absolute number of monocytes; NLR, neutrophils/lymphocytes ratio; NMR, neutrophils/monocytes ratio; LMR, lymphocytes/monocytes ratio; PLT, platelets; Neu, neutrophils; Lym, lymphocytes; Mon, monocytes; CRP, C-reactive protein; PCT, procalcitonin. Statistical significance was calculated according to nonparametric Mann–Whitney test and Fisher’s exact test. Bold values denote statistical significance at the *p* < 0.05 level.

**Table 3 diagnostics-14-00340-t003:** Univariate and multivariate logistic regression analysis for sepsis (without and with CRP).

Predictor	Univariate LR	Multivariate LR without CRP	Multivariate LR with CRP
**Age**	**<0.001**	0.608	0.579
**Sex**	0.155		
**Hb**	**<0.001**	0.797	0.574
**RDW**	**<0.001**	**0.005**	**0.002**
**WBC**	0.490		
**NE#**	0.749		
**LY#**	0.397		
**MO#**	**<0.001**	0.125	**0.026**
**NLR**	0.152		
**NMR**	**0.003**	0.103	0.142
**LMR**	0.295		
**PLT**	0.915		
**NeuX**	**<0.001**	**<0.001**	**0.001**
**NeuY**	**<0.001**	**<0.001**	**0.006**
**NeuZ**	**<0.001**	**<0.001**	**<0.001**
**LymX**	**0.001**	0.648	0.719
**LymY**	0.072		
**LymZ**	0.426		
**MonX**	**<0.001**	**0.040**	0.229
**MonY**	**<0.001**	0.638	0.584
**MonZ**	**<0.001**	**0.031**	**0.005**
**CRP**	**<0.001**		**<0.001**

Hb, hemoglobin; RDW, red distribution width; WBC, white blood cells; NE#, absolute number of neutrophils; LY#, absolute number of lymphocytes; MO#, absolute number of monocytes; NLR, neutrophils/lymphocytes ratio; NMR, neutrophils/monocytes ratio; LMR, lymphocytes/monocytes ratio; PLT, platelets; Neu, neutrophils; Lym, lymphocytes; Mon, monocytes; CRP, C-reactive protein. Bold values denote statistical significance at the *p* < 0.05 level.

**Table 4 diagnostics-14-00340-t004:** AUC, 95% CI, CUT-OFF, sensitivity, specificity for each biomarker (CRP, PCT and multivariate model).

Biomarker	AUC	95% CI	CUT-OFF *	Sensitivity	Specificity
**CRP**	0.83	0.79–0.88	6.07	77%	77%
**PCT**	0.78	0.73–0.84	0.33	77%	70%
**Multivariate Model**	0.92	0.89–0.95	0.655 #	82%	89%

* calculated by Youden index; # cut-off for predicted probabilities of the multivariate model.

**Table 5 diagnostics-14-00340-t005:** Univariate and multivariate Cox regression (CR) analysis for mortality within ICU and hospital.

Predictor	Univariate CR within ICU	Multivariate CR within ICU	Univariate CR within Hospital	Multivariate CR within Hospital
**Age**	**<0.001**	**<0.001**	**<0.001**	**<0.001**
**Sex**	0.370		0.740	
**Hb**	0.087		0.216	
**RDW**	**<0.001**	0.068	**<0.001**	0.138
**WBC**	0.778		0.143	
**NE#**	0.695		0.190	
**LY#**	0.498		0.531	
**MO#**	0.265		0.106	
**NLR**	0.164		**0.027**	0.158
**NMR**	0.536		0.981	
**LMR**	0.210		0.530	
**PLT**	0.522		0.071	
**NeuX**	**0.005**	0.331	**0.005**	0.748
**NeuY**	**<0.001**	**<0.001**	**<0.001**	**0.040**
**NeuZ**	0.258		0.300	
**LymX**	0.657		0.439	
**LymY**	**<** **0.001**	**<0.001**	**<** **0.001**	**<0.001**
**LymZ**	**0.026**	0.320	**0.013**	0.065
**MonX**	**<0.001**	**0.041**	**0.002**	0.373
**MonY**	0.055		**0.024**	**0.021**
**MonZ**	0.146		0.216	
**CRP**	0.292		0.407	

Hb, hemoglobin; RDW, red distribution width; WBC, white blood cells; NE#, absolute number of neutrophils; LY#, absolute number of lymphocytes; MO#, absolute number of monocytes; NLR, neutrophils/lymphocytes ratio; NMR, neutrophils/monocytes ratio; LMR, lymphocytes/monocytes ratio; PLT, platelets; Neu, neutrophils; Lym, lymphocytes; Mon, monocytes. Bold values denote statistical significance at the *p* < 0.05 level.

**Table 6 diagnostics-14-00340-t006:** Hematologic and biochemical data of the study population divided in sepsis and urinary tract infection groups.

Predictor	Without Sepsis (*n* = 56)	With Sepsis (*n* = 223)	*p*-Value
**Hb, g/L**	125 (115–137)	108 (90–128)	**<0.001**
**RDW, %**	14.5 (13.3–15.6)	14.8 (13.8–16.6)	0.074
**WBC, ×10^9^/L**	12.5 (8.5–17.6)	11.2 (6.8–15.8)	0.096
**NE#, ×10^9^/L**	10.4 (7.1–15.3)	9.6 (5.5–14.0)	0.340
**LY#, ×10^9^/L**	1.2 (0.8–1.7)	0.6 (0.4–1.1)	**<0.001**
**MO#, ×10^9^/L**	0.7 (0.5–0.9)	0.4 (0.2–0.7)	**<0.001**
**NLR**	9.1 (4.8–16.6)	13.4 (6.9–24.5)	**0.003**
**NMR**	15.3 (10.7–20.9)	21.5 (13.3–36.2)	**<0.001**
**LMR**	1.6 (1.0–2.5)	1.5 (0.9–2.8)	0.953
**PLT, ×10^9^/L**	264 (200–322)	190 (116–261)	**<0.001**
**NeuX**	387 (350–421)	408 (371–446)	**0.008**
**NeuY**	467 (437–512)	541 (495–607)	**<0.001**
**NeuZ**	1770 (1684–1861)	1792 (1712–1874)	0.084
**LymX**	95 (91–101)	97 (91–104)	0.180
**LymY**	771 (738–823)	775 (728–833)	0.857
**LymZ**	966 (950–988)	954 (931–982)	**0.008**
**MonX**	219 (206–229)	224 (211–245)	**0.003**
**MonY**	1096 (1045–1161)	1144 (1065–1225)	**0.005**
**MonZ**	1349 (1318–1400)	1348 (1303–1408)	0.585
**CRP, mg/L**	94.5 (44.6–154.2)	140.6 (64.8–207.2)	**0.008**
**PCT, ng/mL**	0.37 (0.10–1.74)	2.67 (0.36–19.60)	**<0.001**

Hb, hemoglobin; RDW, red distribution width; WBC, white blood cells; NE#, absolute number of neutrophils; LY#, absolute number of lymphocytes; MO#, absolute number of monocytes; NLR, neutrophils/lymphocytes ratio; NMR, neutrophils/monocytes ratio; LMR, lymphocytes/monocytes ratio; PLT, platelets; Neu, neutrophils; Lym, lymphocytes; Mon, monocytes; CRP, C-reactive protein; PCT, procalcitonin. Statistical significance was calculated according to nonparametric Mann–Whitney test and Fisher’s exact test. Bold values denote statistical significance at the *p* < 0.05 level.

**Table 7 diagnostics-14-00340-t007:** Univariate and multivariate logistic regression analysis for sepsis (without and with CRP) in sepsis and urinary tract infection groups.

Predictor	Univariate LR	Multivariate LR without CRP	Multivariate LR with CRP
**Age**	**<0.001**	**<0.001**	**<0.001**
**Sex**	0.110		
**Hb**	**<0.001**	**0.005**	**0.006**
**RDW**	0.086		
**WBC**	0.510		
**NE#**	0.938		
**LY#**	0.101		
**MO#**	**<0.001**	0.779	0.838
**NLR**	**0.007**	0.180	0.188
**NMR**	**<0.001**	**0.049**	**0.047**
**LMR**	0.322		
**PLT**	**<0.001**	**0.007**	**0.007**
**NeuX**	**0.005**	0.495	0.473
**NeuY**	**<0.001**	**<0.001**	**<0.001**
**NeuZ**	0.061		
**LymX**	0.055		
**LymY**	0.576		
**LymZ**	0.117		
**MonX**	**0.002**	0.916	0.846
**MonY**	**0.007**	**0.013**	**0.013**
**MonZ**	0.845		
**CRP**	**0.017**		0.633

Hb, hemoglobin; RDW, red distribution width; WBC, white blood cells; NE#, absolute number of neutrophils; LY#, absolute number of lymphocytes; MO#, absolute number of monocytes; NLR, neutrophils/lymphocytes ratio; NMR, neutrophils/monocytes ratio; LMR, lymphocytes/monocytes ratio; PLT, platelets; Neu, neutrophils; Lym, lymphocytes; Mon, monocytes; CRP, C-reactive protein. Bold values denote statistical significance at the *p* < 0.05 level.

## Data Availability

The original contributions presented in the study are included in the article, further inquiries can be directed to the corresponding author.

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
