# Peer review of "The Role of New Morphological Parameters Provided by the BC 6800 Plus Analyzer in the Early Diagnosis of Sepsis"

_diagnostics, 2024, doi:10.3390/diagnostics14030340_

Round 1

Reviewer 1 Report

Comments and Suggestions for Authors

In the paper” The role of new morphological parameters provided by the BC 6800 plus analyzer in the early diagnosis of sepsis”, authored with precision and insight, the article sheds light on the innovative use of the BC 6800 Plus Analyzer in enhancing the early detection of sepsis.

This is not necessarily a new subject, but I think that the novelty resides in the fact that the assessment of novel parameters (Neu-Z, Mon-Z) can be useful to diagnose early septic patients. Also, established of a new model which include many parameters for early diagnose of sepsis.

This review aims to highlight the significance of the findings and applaud the authors for their valuable insights.

The core strength of the article lies in its meticulous exploration of the BC 6800 Plus Analyzer's new morphological parameters and their application in diagnosing sepsis at an early stage. The authors delve into the specifics of these parameters, elucidating how they provide a more nuanced understanding of the morphological changes in blood cells associated with sepsis. This level of detail is crucial for clinicians seeking precise and timely diagnostic tools.

One key aspect of the article's merit is its emphasis on the clinical relevance of the BC 6800 Plus Analyzer's morphological parameters. The authors present the technical details of the analyzer and also connect the dots to the real-world implications for patients. By doing so, they made the article accessible and impactful for a broad readership, including clinicians, researchers, and healthcare administrators.

The inclusion of case studies and clinical examples further strengthens the article's credibility.

The article's clear and concise writing style deserves acknowledgment. Complex scientific concepts are presented in an accessible manner, allowing readers from diverse backgrounds to grasp the intricacies of the BC 6800 Plus Analyzer's morphological parameters and their relevance in sepsis diagnosis.

In conclusion, the paper stands as a commendable work that advances our understanding of sepsis diagnosis. The article's meticulous exploration of the BC 6800 Plus Analyzer's morphological parameters, coupled with its clinical relevance and clear presentation, positions it as a significant contribution to the medical literature.

The references were recently and carefully selected, and shows the accuracy of the information presented in the study by the authors.

The article has same minor limitations, such as methods section should include detailed characteristics of morphological patterns of neutrophils, monocytes, lymphocytes and erythrocytes to identify sepsis (cellular pattern X, Y, Z). The discussion section should have detailed description of all cellular pattern X, Y, Z because all morphological patterns of neutrophils, monocytes, lymphocytes and erythrocytes are involved to early diagnose sepsis.

The manuscript is overall well-written with minor style errors that need some editing after careful revision.

I therefore recommend that the paper can be published after corrections to minor methodological errors and text editing.

Reviewer 2 Report

Comments and Suggestions for Authors

The paper represents a comprehensive study describing significant experience (with a specimens from a total of 527 patients) on the use of automatically detected various morphological patterns of neutrophils, monocytes, lymphocytes and erythrocytes to identify sepsis patients vs infected and non-infected cohorts with no sepsis. The paper significantly extends the study by other authors (Zhang W, Zhang Z, Pan S. et al., 2021) who have been first to describe similar technological approach  and parameters for sepsis diagnosis in a cohort with an ethnically different background. The main novelty and advantage of the reviewed paper include inclusion novel parameters (Neu-Z, Mon-Z) that were informative to determine sepsis, and demonstration that a combination of several morphological parameters may diagnose sepsis at a highest AUC value (0,92) in a multivariate model. Several limitations may easily be repaired to improve scientific soundness and quality of presentation: (1) characteristics of expected differential contribution of cellular complexity, nucleic acid content etc. to each of the cellular pattern X,Y,Z should be explained in the Methods section; (2) the Discussion section seems too brief since  the X pattern is mainly discussed while the origin of Y and Z parameters and  their contributions to sepsis are discussed too briefly; this is not fair because  the Y and Z patterns of monocytes and neutrophiles exhibit significant  contribution to the diagnostic informativity of cell morphology testing  by hematology analyzer in both univariate and multivariate formats.

Reviewer 3 Report

Comments and Suggestions for Authors

The authors deal with a very interesting and very modern topic. Research on the early diagnosis of sepsis and septic shock is producing many papers, all very interesting.

The paper is well written, the statistical analysis is very good and it has been articulated in a very effective way.

However, the paper has some biases.

The evaluation of leukocyte morphological parameters highlights the role of innate immunity and its dysregulation in the pathophysiology of sepsis.

But the authors do not address this issue even in "Discussion".

It would be useful to add a paragraph in "Discussion" on the role of innate immunity and early markers of the alteration (HMGB-1, Mertalloproteinase etc).

Why not use such very early biomarkers of sepsis and sepsis-related AKI?

The authors could explain it, adding another paragraph in "Discussion" and the related references in this sense (Campo S et al, Lacquaniti A. et al etc)

Round 2

Reviewer 3 Report

Comments and Suggestions for Authors

The authors made the requested changes